# Controlled Release of Bone Morphogenetic Protein-2 Augments the Coupling of Angiogenesis and Osteogenesis for Accelerating Mandibular Defect Repair

**DOI:** 10.3390/pharmaceutics14112397

**Published:** 2022-11-07

**Authors:** Hao Yao, Jiaxin Guo, Wangyong Zhu, Yuxiong Su, Wenxue Tong, Lizhen Zheng, Liang Chang, Xinluan Wang, Yuxiao Lai, Ling Qin, Jiankun Xu

**Affiliations:** 1Musculoskeletal Research Laboratory and Centre of Musculoskeletal Aging and Regeneration, Department of Orthopaedics and Traumatology, Faculty of Medicine, The Chinese University of Hong Kong, Hong Kong SAR, China; 2Innovative Orthopaedic Biomaterial and Drug Translational Research Laboratory, Li Ka Shing Institute of Health, The Chinese University of Hong Kong, Hong Kong SAR, China; 3Division of Oral and Maxillofacial Surgery, Faculty of Dentistry, The University of Hong Kong, Hong Kong SAR, China; 4Translational Medicine R&D Center, Institute of Biomedical and Health Engineering, Shenzhen Institutes of Advanced Technology, Chinese Academy of Sciences, Shenzhen 518057, China; 5Joint Laboratory of Chinese Academic of Science and Hong Kong for Biomaterials, The Chinese University of Hong Kong, Hong Kong SAR, China

**Keywords:** bone morphogenetic protein-2, hydrogel, mandibular defect, osteogenesis, angiogenesis

## Abstract

Reconstruction of a mandibular defect is challenging, with high expectations for both functional and esthetic results. Bone morphogenetic protein-2 (BMP-2) is an essential growth factor in osteogenesis, but the efficacy of the BMP-2-based strategy on the bone regeneration of mandibular defects has not been well-investigated. In addition, the underlying mechanisms of BMP-2 that drives the bone formation in mandibular defects remain to be clarified. Here, we utilized BMP-2-loaded hydrogel to augment bone formation in a critical-size mandibular defect model in rats. We found that implantation of BMP-2-loaded hydrogel significantly promoted intramembranous ossification within the defect. The region with new bone triggered by BMP-2 harbored abundant CD31+ endomucin+ type H vessels and associated osterix (Osx)+ osteoprogenitor cells. Intriguingly, the new bone comprised large numbers of skeletal stem cells (SSCs) (CD51+ CD200+) and their multi-potent descendants (CD51+ CD105+), which were mainly distributed adjacent to the invaded blood vessels, after implantation of the BMP-2-loaded hydrogel. Meanwhile, BMP-2 further elevated the fraction of CD51+ CD105+ SSC descendants. Overall, the evidence indicates that BMP-2 may recapitulate a close interaction between functional vessels and SSCs. We conclude that BMP-2 augmented coupling of angiogenesis and osteogenesis in a novel and indispensable way to improve bone regeneration in mandibular defects, and warrants clinical investigation and application.

## 1. Introduction

Mandibular defects lead to facial deformities and dysfunctions, causing sociopsychological impacts and severely affecting quality of life [1,2]. Tumor resections, trauma, and inflammatory and infectious diseases can result in major mandibular defects [1,2]. Mandibular reconstruction is challenging, from both functional and esthetic points of view. The current standardized strategy for mandibular defect reconstruction is osseous free flap transplantation [1,3]. However, there have been concerns regarding donor-site morbidity and limited amounts of available bone [4,5]. Thus, the development of novel therapeutic alternatives is needed.

During the embryonic developmental stage, the formation of mandibular bone mainly undergoes intramembranous ossification [6,7]. Intriguingly, enhanced intramembranous ossification contributes to gradual distraction-induced bone formations in the mandible [8]. Thus, intramembranous ossification plays an essential role in bone regeneration of the mandible. Multiple cell types including endothelial and osteogenic cells are recruited to initiate angiogenesis and osteogenesis, followed by intramembranous ossification. Previous studies have demonstrated a type of functional blood vessel, called the type H vessel, that shows high expression of both CD31 and endomucin (Emcn) and is crucial for bone development and regeneration by supporting perivascular osteoprogenitor cells [9]. Meanwhile, a recent study found that type H vessels were also distributed at the mandibular condyle and strongly correlated with bone formation [10]. In addition, skeletal stem cells (SSCs) have been isolated and their indispensable roles in skeletal tissues, including mandible regeneration, are well-validated [8,11,12]. SSCs can differentiate into multi-potent descendants, such as bone cartilage skeletal progenitors (BCSPs) and osteoprogenitors (Ops) that mainly contribute to the massive bone regeneration induced by mandibular distraction [8]. Therefore, new strategies aimed at facilitating the growth of type H vessels and SSCs may trigger more bone regeneration in mandibular defects.

The bone morphogenetic protein-2 (BMP-2) is a growth factor important for osteogenesis. Recombinant human BMP-2 is currently used in orthopaedics, such as spinal fusion [13,14,15,16]. However, efficacy of a BMP-2-based strategy for bone regeneration of mandibular defects has not been fully investigated. One previous study reported that BMP-2 promoted a coupling growth of type H vessels and multipotent stromal cells, and induced heterotopic ossification in the tendons of mice [17]. Moreover, recent evidence has shown that BMP-2 can inspire periosteum-like tissue, which harbors abundant periosteum-derived stromal cells coupled with type H vessels for bone regeneration of calvarial defects [17]. BMP-2 has been demonstrated as a critical growth factor that stimulates the expansion, self-renewal, and differentiation of SSCs [11]. Based on these findings, we hypothesized that BMP-2 could enhance the coupling of angiogenesis and osteogenesis to augment mandibular regeneration. In addition, due to advantages in biocompatibility, controllable degradability, and easy modification, a hydrogel was used as the delivery system for BMP-2 in mandibular reconstruction [18].

In this study, we fabricated a BMP-2-loaded hydrogel and implanted it to enhance bone regeneration in a critical-size mandibular defect model in rats. We found that the released BMP-2 significantly promoted new bone formation in the mandibular defect model via augmentation of intramembranous ossification invaded by abundant type H vessels and SSCs. This BMP-2-based strategy warrants further clinical investigation before bedside application.

## 2. Materials and Methods

### 2.1. Fabrication of BMP-2-Loaded Hydrogel

A BMP-2-loaded hydrogel was fabricated by encapsulating BMP-2 with a gelatin methacryloyl (GelMA) hydrogel (Figure 1A). GelMA was synthesized as previously described (Figure 1A) [19,20]. Briefly, gelatin (Type B, Sigma Aldrich, St. Louis, MO, USA) was dissolved in a carbonate-bicarbonate (CB) buffer (0.25 M) at 10% *w*/*v* before heating to 55 °C. Then, the pH of the gelatin solution was adjusted to 9.4. Methacrylic anhydride (Sigma Aldrich, St. Louis, MO, USA, 94%) was added dropwise to the gelatin solution with continuing stirring. The reaction proceeded for 1 h at 55 °C, and the final pH of the reaction solution was adjusted to 7.4 to stop the reaction. The solution was dialyzed using a membrane with a molecular weight cut-off (MWCO) of 12,000–14,000 Da against deionized water (ddH_2_O) at 37 °C for 7 days, The solution was then filtered through a 0.22 μm membrane, and frozen and lyophilized until dry. The resultant GelMA product was stored in the dark at −20 °C. The GelMA solution was prepared at the indicated concentrations of 8% *w*/*v* by dissolving it in phosphate buffered saline (PBS, pH 7.4). BMP-2 was dissolved at a concentration of 50 μg/mL in the GelMA solution. The hydrogel was synthesized by mixing the above solution with the photo-initiator, 2-hydroxy-4′-(2-hydroxyethoxy)-2-methylpropiophenone (Irgacure 2959, Ciba, Basel, Switzerland), at a final concentration of 0.05% *w*/*v*, followed by exposure to ultraviolet light (UV, 95 mW/cm^2^) for 6 min. The final 40 μL BMP-2-loaded hydrogels (net 2 μg of BMP-2 per implant) were prepared for implantation.

### 2.2. Scanning Electron Microscopy (SEM)

The microstructural characterization of different hydrogels was evaluated using scanning electron microscopy (Quanta-400, FEI, Hillsboro, OR, USA) at an accelerating voltage of 10 kV. Hydrogel samples were frozen, lyophilized until dry, and coated with gold powder before observation. Cross-sectional images were obtained.

### 2.3. Degradation Test In Vitro

To investigate the degradation of the GelMA hydrogel and BMP-2-loaded hydrogel, all samples were soaked in PBS solution at 37 °C in a shaker. The ratio of the PBS volume to the hydrogel was set to 1 mL:100 mg. At days 1, 3, 7, 10, 14, and 21, the samples were lyophilized for 24 h and weighed.

### 2.4. Swelling Test In Vitro

The swelling property of hydrogel samples was evaluated by the weighing method. Briefly, all hydrogel samples were lyophilized to obtain their dry weight. Then, the hydrogels were immersed in 10 times their volume in PBS at 37 °C for 1, 3, 5, and 10 h. Thereafter, the swelling weights (wet weight) were measured. The swelling ratio was calculated based on the formula: (wet weight − dry weight)/dry weight.

### 2.5. Establishment of Critical-Size Mandibular Defect Model in Rats

The critical-size mandibular defect model in rats was established based on previous descriptions with optimization [21,22], after obtaining ethical approval from the animal experiment ethics committee of the Chinese University of Hong Kong. Female SD rats (3 months old) were used. After deep anaesthesia by intra-peritoneal (i.p.) injection of a mixture of xylazine, ketamine, and saline (ratio 2:3:3, at a dose of 0.2 mL/100 g body weight). An incision overlying and parallel to the left mandible was made to expose the angle of the mandible. A circular defect was created using a 4 mm circular drill at the mandible. Bone dusts were removed by saline irrigation. Thereafter, the wounds were directly closed for rats in the control group (Ctrl, *n* = 3–4 per time points). A GelMA hydrogel or BMP-2-loaded GelMA hydrogel was implanted into the defect for rats in the hydrogel (Gel, *n* = 4 per time points) or BMP-2-loaded hydrogel (BMP-2 Gel, *n* = 4 per time points) group, respectively. Lastly, the wound was closed. The rats were allowed to recover on a heating pad. After surgery, i.p. injection of temgesic (0.05 mg/kg of body weight) were conducted twice daily for 2 days to minimize pain for the rats. At weeks 2 and 8, the rats were euthanized by a lethal dose of pentobarbital (25%) before samples were harvested.

### 2.6. Micro-CT (μCT) Scanning

The fixed mandible samples were scanned by μCT (μCT 40, SCANCO MEDICAL, Brttisellen, Switzerland; voltage = 70 kV; current = 113 μA; voxel size = 15 μm; threshold = 200). The microarchitecture of the newly generated bone (defined as a 5 × 5 × 1.5 mm^3^ volume) in the mandibular defect was evaluated. Bone volume/total tissue volume (BV/TV), trabecular thickness (Tb.Th), bone mineral density (BMD), and trabecular number (Tb.N) were calculated [23].

### 2.7. Histomorphometric Analysis

The mandibles were fixed in 4% PFA for 2 days, decalcified in 9% formic acid for 7 days before paraffin embedding. Sections 5 μm thick were prepared. For H&E staining [24,25], after deparaffinization and hydration, the sections were stained with hematoxylin for 10 min and eosin for 2 min. Then, the sections were mounted with DPX (Sigma, #44581). Trichrome staining was performed using the Trichrome Staining Kit (abcam, Cambridge, UK ab150686), according to the manufacturer’s instructions [26]. The interface of new bone to old bone was identified, and the area of new bone and old bone were measured using ImageJ, based on the H&E slides. The ratio of new bone to old bone was calculated.

### 2.8. Immunofluorescent (IF) Staining

IF staining was performed according to our previous protocols [24,26,27]. After deparaffinization and hydration, the sections were subjected to a heat-induced antigen retrieval using 10 mM citrate buffer for 20 min. Thereafter, the sections were blocked in 1*PBST, supplemented with 0.3% Triton X-100, 1% BSA, and 2% FBS for 30 min at room temperature, followed by incubation with the appropriate dilution of primary antibodies (Appendix A), including anti-CD31, anti-Emcn, anti-Osx, anti-CD51, anti-CD200, and anti-CD105, in 1*PBST supplemented with 0.05% Triton X-100, 1% BSA, and 2% FBS, overnight, at 4 °C. The sections were washed three times and incubated with the appropriate dilution of secondary antibodies (Appendix A), including Alexa Fluor 488, 546, or 647 conjugated anti-rat, anti-mouse, anti-rabbit, or anti-goat antibodies for 1 h at room temperature. The sections were washed three times before being mounted by DAPI (ThermoFisher Scientific, P36934, Waltham, MA, USA). Three to four images were captured from each section. The fraction of CD31+ Emcn+ to the fields, as well as the percentage of either CD51+, Osx+, and CD51+ CD200 or CD51+ CD105 to total cells in the fields were calculated and averaged.

### 2.9. Statistical Analysis

Sample size was estimated by power calculations based on our pilot study. Four rats per group were sufficient to detect a 30% difference in the ratio of new bone to old bone between the hydrogel and BMP-2-loaded hydrogel group at week 2 post-implantation. The histological semi-quantitative analysis was performed on three sections, with an interval of 100 μm. The average was used for subsequent statistical analysis. Histological semi-quantitative analysis and μCT data evaluation were performed by two colleagues in a blinded fashion. In cases of discrepancies, opinions from a third co-investigator were sought. Differences among the groups were analyzed using one-way ANOVA and Dunn’s multiple comparisons test using GraphPad Prism software (Version 8.2). Statistical significance was defined as *p* < 0.05.

## 3. Results

### 3.1. Characterization of BMP-2-Loaded Hydrogel

The in vitro degradation pattern of the GelMA hydrogel and BMP-2-loaded GelMA hydrogel were evaluated by soaking the samples in PBS. The results showed no difference in the degradation rate between groups (Figure 1B). The weight loss of the GelMA hydrogel and BMP-2-loaded GelMA hydrogel reached 74.1 ± 4.6 and 76.6 ± 4.0% within 21 days (Figure 1B), respectively. Both hydrogels were fully swelled in PBS at 37 °C for 10 h. The final mass swelling ratios of the GelMA hydrogels and BMP-2-loaded GelMA hydrogels were 734.6 ± 5.5 and 731.3 ± 11.4%, respectively (Figure 1C). They all exhibited good hydrophilicity and water absorption, and could recover its initial shape in a short time. The internal porous structure and morphology of hydrogels were evaluated by SEM. The hydrogels had a randomly oriented porous structure with a diameter of 30–100 μm (Figure 1D). BMP-2 uniformly attached to the hydrogel networks in a filamentous-like structure (Figure 1D). Although the inner wall of the hydrogel shrank and collapsed after loading BMP-2, BMP-2-loaded hydrogels still possessed a porous structure (Figure 1D). As reported, such structures of hydrogels provided advantages for protein adsorption, cell adhesion, and infiltration [28].

### 3.2. Implantation of BMP-2-Loaded Hydrogel Promotes Bone Regeneration of Mandibular Defect

To evaluate the regenerative efficacy of the BMP-2-loaded hydrogel in the mandible, we established a critical-size mandibular defect model in rats (Figure 2A). Histological analysis showed that only a small amount of new bone formed surrounding the old bone at the defect sides in the blank control group at week 2, indicating weak intramembranous ossification (Figure 2B). Similar results were found in the group treated by hydrogel without BMP-2 (Figure 2B). In contrast, the BMP-2-loaded hydrogel significantly enhanced intramembranous ossification and enlarged the formation of new bone. The ratio of new bone to old bone (NB/OB) also dramatically increased compared with either the control or hydrogel group (Figure 2B,C). At week 8, there was no evident increase in new bone formation at the defect sides in both control and hydrogel groups (Figure 2B). In the BMP-2-loaded hydrogel group, new bone was continuously formed toward the center of the defect site. We further observed that previously formed bony tissue was gradually flattened, and the thickness of trabecular bone within the new bone was increased in all the groups (Figure 2B). The NB/OB ratio in the BMP-2-loaded hydrogel group was slightly decreased at week 8, relative to that of week 2, which may have resulted from remodeling; the BMP-2-loaded hydrogel group still showed the highest NB/OB ratio among all the groups at week 8 post-operation (Figure 2B,C).

μCT scanning showed that the BV/TV ratio in the BMP-2-loaded hydrogel group was significantly higher than that of the control group at both week 2 and 8 (Figure 2D,E). Implantation of the hydrogel alone only slightly increased the BV/TV ratio (Figure 2D,E). The Tb.N was markedly increased in the BMP-2-loaded hydrogel group at week 2, whereas the difference in Tb.N among the groups was not significant at week 8, which may have been due to the gradually increased Tb.Th at this time point (Figure 2D,E). In addition, the BMD showed no significant difference among the groups (Figure 2D,E).

### 3.3. BMP-2 Induces Angiogenesis and Growth of Functional Vessels

Functional vessels and associated stem or progenitor cells are responsible for initiation of intramembranous ossification [29,30,31]. Type H vessels expressing high levels of CD31 and Emcn serve as one of the key functional vessel subtypes that drive the coupling of angiogenesis and osteogenesis [9,10]. In this study, we performed immunofluorescence (IF) to determine the distribution of type H vessels in the mandibular defect model. We found that type H vessels were mainly distributed at the interface of new bone to defect and the surface of trabecular bone of the newly regenerated bone (Figure 3A, Appendix A). At week 2, there was limited growth of type H vessels in the new bone of the control group; implantation of the hydrogel alone did not promote the growth of type H vessels in the new bone relative to the control (Figure 3A, Appendix A). We found that the BMP-2-loaded hydrogel dramatically induced the invasion of type H vessels, especially at the interface of new bone to defect (Figure 3A, Appendix A). The area fraction of CD31+ Emcn+ to new bone significantly increased in the BMP-2-loaded hydrogel group relative to the control and hydrogel only groups (Figure 3B). At week 8, the growth of type H vessels in the new bone was markedly reduced among all the groups, with a decreased area fraction of CD31+ Emcn+ to new bone (Figure 3A,B, Appendix A). There was only scattered CD31+ Emcn+ vessels in the new bone of the control and hydrogel only groups (Figure 3A, Appendix A). Although the invasion of type H vessels was reduced, the area fraction of CD31+ Emcn+ to new bone in the BMP-2-loaded hydrogel group was still significantly higher than those in other groups (Figure 3A,B, Appendix A).

### 3.4. BMP-2 Enhances the Coupling of Angiogenesis and Osteogenesis

To further confirm the functional vessels invaded in the new bone and investigate the coupling of angiogenesis and osteogenesis with or without implantation of the BMP-2-loaded hydrogel, we co-stained Emcn and osterix (Osx) in the newly formed bone using IF staining. We identified that the Osx+ osteoprogenitor cells were mainly distributed adjacent to the Emcn+ vessels (Figure 4A, Appendix A). At week 2 post-operation, there was no difference between control and hydrogel groups in terms of Osx+ osteoprogenitor cell infiltration in the new bone (Figure 4A, Appendix A). In the BMP-2-loaded hydrogel group, the regenerated bone harbored more abundant Emcn+ vessels and associated Osx+ osteoprogenitor cells than those in the control and hydrogel groups (Figure 4A, Appendix A). Quantitative data based on the IF staining further showed that the percentage of Osx+ cells to total cells in the new bone was significantly elevated in the BMP-2-loaded hydrogel group compared with the control and hydrogel group, which indicated an enhanced coupling of angiogenesis and osteogenesis induced by BMP-2 (Figure 4A,B, Appendix A). At week 8, with the declining vascularization, the number of Osx+ osteoprogenitor cells in the newly generated bone decreased among all the groups; however, the BMP-2-loaded hydrogel group had relatively higher numbers of Osx+ osteoprogenitor cells (Figure 4A,B, Appendix A).

### 3.5. BMP-2 Recruits Abundant Skeletal Stem Cells

CD51 has been identified as a relative specific marker expressed in both SSCs and their multi-potent progeny [8,11,12]. In this study, we measured the distribution and frequency of CD51+ cells in the newly regenerated surface of the trabecular bone and interface of bone to defect where abundant vasculature (region of Emcn+) was observed (Figure 5A, Appendix A). At week 2, we found limited numbers of CD51+ cells in the newly generated bone in both the control and hydrogel groups (Figure 5A, Appendix A). After implantation of the BMP-2-loaded hydrogel, the regenerated bone harbored larger numbers of CD51+ cells (Figure 5A, Appendix A). Quantitative data further showed that the percentage of CD51+ cells to total cells was significantly increased in the BMP-2-loaded hydrogel group compared with the other groups (Figure 5A,B, Appendix A). Moreover, we found that multiple CD51+ cells always adhered together and formed cell clusters at week 2 after implantation of the BMP-2-loaded hydrogel, indicating enhanced cell expansion (Figure 5A, Appendix A). At week 8, the number of CD51+ cells declined in all groups, indicating weak anabolism of bone at the late healing stage (Figure 5A,B, Appendix A).

Previous studies reported that SSCs (CD51+ CD200+) could further differentiate into their multi-potent descendants, BCSPs (CD51+ CD105+), chondroprogenitors (OPs) (CD51+ CD90+ CD105+ CD200+), and OPs (CD51+ CD90+ CD105+), which subsequently participated in tissue regeneration [8,11]. In this study, we determined the fraction of CD51+ CD200+ SSCs and their CD51+ CD105+ descendants in total CD51+ cells using IF staining. The results showed that the percentage of CD51+ CD200+ SSCs to CD 51+ cells reached around 30% in the control and hydrogel groups at week 2 post-operation; the percentage of CD51+ CD200 SSCs slightly increased up to around 40% in the BMP-2-loaded hydrogel group (Figure 6A,C, Appendix A). At week 8, the proportion of CD51+ CD200+ SSCs in CD51+ cells remained around 30% among the groups (Figure 6A,C, Appendix A). At week 2 post-operation, the percentages of CD51+ CD105+ descendants in total CD51 cells were around 30% in both control and hydrogel groups (Figure 6B,D, Appendix A). Intriguingly, the number of CD51+ CD105+ descendants was significantly increased after implantation of the BMP-2-loaded hydrogel compared with the control and hydrogel groups (Figure 6B,D, Appendix A). Quantitative data showed that the fraction of CD51+ CD105+ cells in total CD51+ cells increased to around 50% in the BMP-2-loaded hydrogel group (Figure 6B,D, Appendix A). At week 8 after implantation, the fractions of CD51+ CD105+ cells were increased to around 40% (around 30% at week 2) in both the control and hydrogel groups (Figure 6B,D, Appendix A). The BMP-2-loaded hydrogel group had a proportion of CD51+ CD105+ descendants of around 50% (Figure 6B,D, Appendix A). The above results indicate that the implanted BMP-2 facilitated the differentiation of SSCs into their multi-potent progenies.

## 4. Discussion

In this study, we developed a BMP-2-loaded hydrogel and investigated its efficacy in promoting bone regeneration after implantation in critical-size mandibular defect. We found that implantation of BMP-2-loaded hydrogel facilitated bone regeneration. BMP-2-induced new bone comprised abundant type H vessels and associated Osx+ osteoprogenitor cells. Furthermore, we observed that BMP-2 recruited more CD51+ SSCs, adjacent to the invaded vessels in the newly regenerated bone. The enhanced coupling of angiogenesis and osteogenesis mediated by BMP-2 contributes to mandibular regeneration.

During embryonic development, the neural crest drives the formation of the mandible [6,8]. The process of mandibular development mainly undergoes intramembranous ossification [6,7]. We confirm that the healing of mandibular defects also depends on periosteum-mediated intramembranous ossification in the animal model of our present study, indicating that the healing of mandibular defects may mimic the process of mandibular development. The BMP-2 signaling pathway is essential in regulating bone formation during mandibular development [32]. In this study, we observed that implantation of a BMP-2-loaded hydrogel stimulated enhanced intramembranous ossification.

Enhanced angiogenesis is indispensable to intramembranous ossification [33]. Type H vessels with high co-expression of CD31 and Emcn was involved in bone regeneration by supporting the perivascular Osx+ osteoprogenitor cells [9]. A recent study confirmed the existence of type H vessels in the mandibular condyle [10]. However, the roles of type H vessels in mandibular defect healing has yet not been determined. In the present study, we found that CD31+ Emcn+ type H vessel invaded into the newly generated bone associated with Osx+ osteoprogenitor cells, indicating that type H vessels indeed participate in the bone regeneration of mandibular defects. In addition, our histological analysis showed that the bone regeneration of mandibular defects was mainly contributed to by intramembranous ossification, which is regulated by the periosteum. As extensive vascularization is a representative feature of the periosteum [31,34,35,36], we speculate that the type H vessels that drive bone regeneration in mandibular defects may originate from the periosteum. The exact distribution and function of type H vessels in the periosteum of the mandible require further investigation. BMP-2 has been demonstrated as a pivotal growth factor for angiogenesis in bone regeneration [37,38]. Previous studies also show that BMP-2 can induce the growth of type H vessels [17,31]. In this study, we found that BMP-2-induced newly formed bone harbored abundant CD31+ Emcn+ type H vessels coupled with a large number of Osx+ osteoprogenitor cells. Our findings demonstrate that implantation of a BMP-2-loaded hydrogel can facilitate type H vessel invasion and associated osteoprogenitor cell infiltration in mandibular defect healing, which is consistent with previous reports [17,31].

A previous study demonstrated that mandibular distraction activated SSCs that contribute to new bone formation, indicating the essential role of SSCs in mandibular regeneration [8]. In the current study, we measured the occurrence of SSCs in a mandibular defect model. We found that SSCs and their descendants were mainly distributed at the surface of trabecular bone and the interface of bone and defect. We also observed that the SSCs were adjacent to blood vessels, indicating that the SSCs may also be maintained by the blood vessels. After implantation of the BMP-2-loaded hydrogel, the SSCs and their descendants significantly increased. Moreover, multiple CD51+ cells adhered together and formed cell clusters, indicating that the released BMP-2 promotes the expansion of SSCs [11,39]. BMP-2 also has the capacity of determining the cell fate of SSCs [11]. Here, we further evaluated the fractions of either CD51+ CD200+ SSC or CD51+ CD105+ SSC descendants. Our results showed that the percentage of CD51+ CD105+ SSC descendants markedly increased after implantation of the BMP-2-loaded hydrogel. These results indicated that the released BMP-2 may promote SSC differentiation into their multi-potent descendants and participate in subsequent bone regeneration. On the other hand, histological analyses showed no cartilage tissue at the defect side, indicating that SSCs may ultimately commit to the OP lineage. Vascular endothelial growth factor (VEGF) is another crucial growth factor that is involved in osteogenesis and manipulating the fate of SSCs [11,12,40,41,42]. VEGF or its antagonist can work together with BMP-2 to promote BCSP differentiation to OPs (BMP-2 + VEGF or VEGF agonist) or CPs (BMP-2 + VEGF antagonist or inhibitor) [11,12]. As we found that the SSCs were mainly distributed adjacent to blood vessels, we speculate that VEGF secreted from these vessels may participate in driving BCSP differentiation to the OP lineage. After implantation of the BMP-2-loaded hydrogel, the released BMP-2 could also stimulate the invasion of blood vessels, which may further support the cell fate determination of associated SSCs. Therefore, hydrogel-released BMP-2 and vessel-secreted VEGF appear to be important mechanisms affecting SSC lineage commitment. However, the exact relationship between functional blood vessels and SSCs during the bone regeneration of mandibular defects requires further clarification in future investigations.

Previous studies have demonstrated that a BMP-2 (50 μg/mL)-loaded hydrogel can offer rapid release of around 60% BMP-2 after the first 24 h, followed by a slower and sustainable release with 0.3–0.5% of the loaded BMP-2 every 2–3 days [43,44]. Our present study found that implantation of a BMP-2-loaded hydrogel dramatically elevated bone formation in the mandibular defect model. However, the mandibular defect in our study was still not completely healed in BMP-2-loaded hydrogel group. Thus, the doses of BMP-2 and release pattern of hydrogel require further modifications for massive bone regeneration of critical-size mandibular defects.

## 5. Conclusions

Our current work provides evidence that implantation of a BMP-2-loaded hydrogel augments the bone regeneration of critical-size mandibular defects in rats. Enhanced coupling of angiogenesis and osteogenesis appears to the key mechanisms underlying the effect of BMP-2 on mandibular regeneration. BMP-2 and hydrogel have been approved for clinical use, and their combined application may favor bioactivity and biocompatibility, two essential elements in regenerative medicine. Herewith, the strategy of the BMP-2-loaded hydrogel may be promising for bone regeneration of mandibular defects, and warrants further clinical investigation before wide clinical application.

## Figures and Tables

**Figure 1 pharmaceutics-14-02397-f001:**
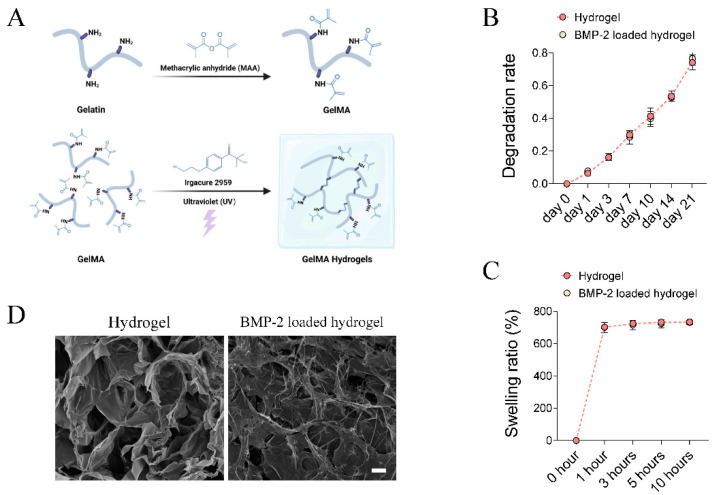
Fabrication and characterization of BMP-2-loaded hydrogel. (**A**) Schematic chart showing the synthesis of BMP-2-loaded GelMA hydrogel. (**B**,**C**) Quantitative analysis of the degradation rate and swelling ratio of GelMA and BMP-2-loaded GelMA hydrogels. (**D**) Representative SEM image of the microstructure of GelMA and BMP-2 loaded GelMA hydrogels. Scale bar 25 μm.

**Figure 2 pharmaceutics-14-02397-f002:**
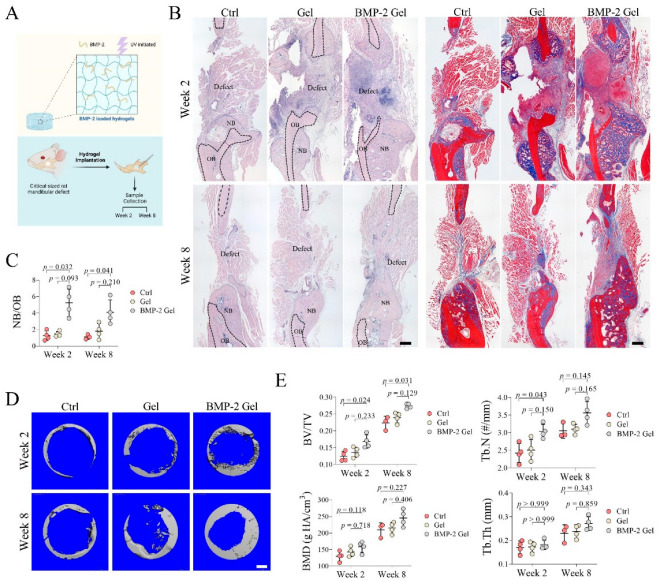
Implantation of BMP-2-loaded hydrogel promoted bone regeneration in mandibular defects of rats. (**A**) Schematic chart of the study design. (**B**) Trichrome staining of mandibular bone. Scale bar, 250 μm. The black dashed line indicates the border of the new bone and old bone. NB: new bone, OB: old bone. (**C**) Quantitative analysis of the ratio of new bone to old bone (NB/OB). (**D**) Reconstructed 3D images of the defect in mandibular bone. Scale bar, 750 μm. (**E**) Quantitative analysis of bone volume/total tissue volume (BV/TV), trabecular number (Tb.N), bone mineral density (BMD), and trabecular thickness (Tb.Th). The *p* value was calculated by one-way ANOVA and Dunn’s multiple comparisons test.

**Figure 3 pharmaceutics-14-02397-f003:**
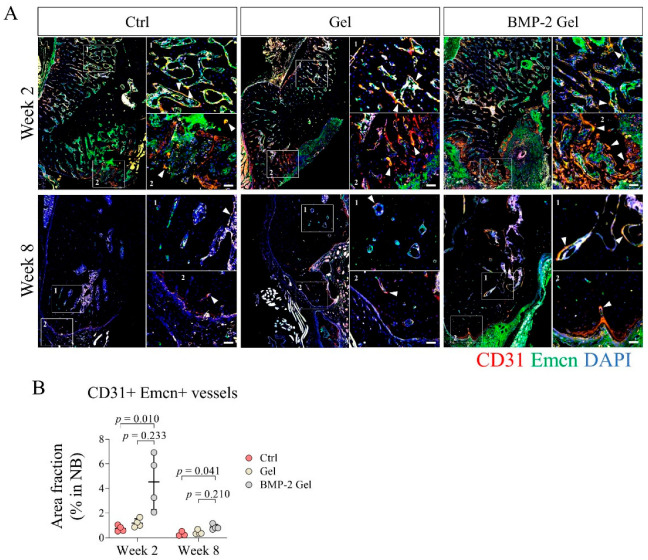
BMP-2 enhanced angiogenesis and growth of type H vessels after implantation during the mandibular defect repair. (**A**) Immunofluorescent staining of type H vessels (CD31+ Emcn+) in the mandibular bone. Scale bar, 50 μm. The white arrow indicates the type H vessels. (**B**) Quantitative analysis of the area fraction of type H vessels in the new bone. The *p* value was calculated by one-way ANOVA and *Dunn’s* multiple comparisons test.

**Figure 4 pharmaceutics-14-02397-f004:**
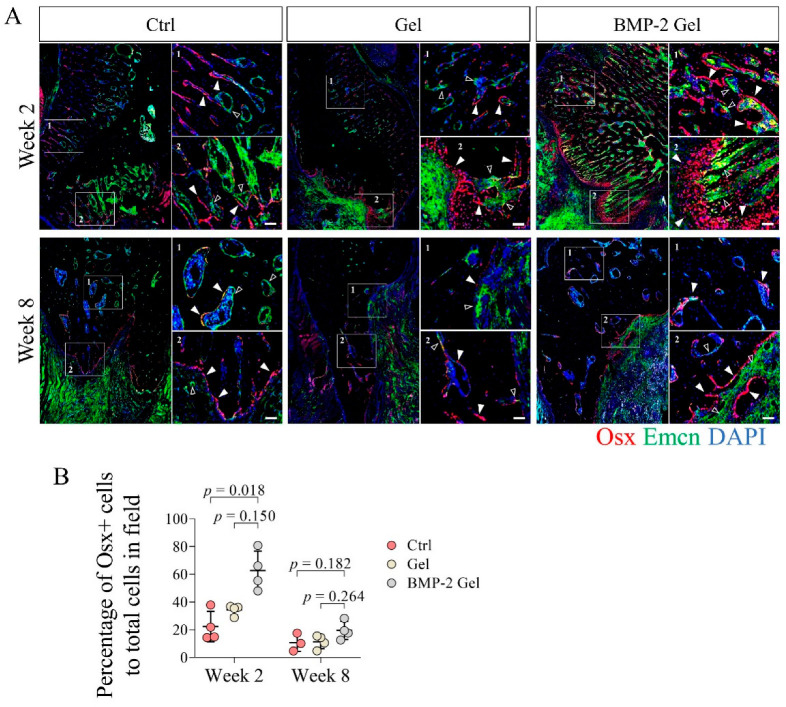
BMP-2 increased the number of osterix (Osx)+ osteoprogenitor cells coupled with vessels after implantation during the mandibular defect repair. (**A**) Immunofluorescent staining of Osx+ osteoprogenitor cells in the mandibular bone. Scale bar, 50 μm. The white arrow indicates the Osx+ osteoprogenitor cells. The hollow arrow indicates the vessels. (**B**) Quantitative analysis of the number of Osx+ osteoprogenitor cells in the new bone. The *p* value was calculated by one-way ANOVA and *Dunn’s* multiple comparisons test.

**Figure 5 pharmaceutics-14-02397-f005:**
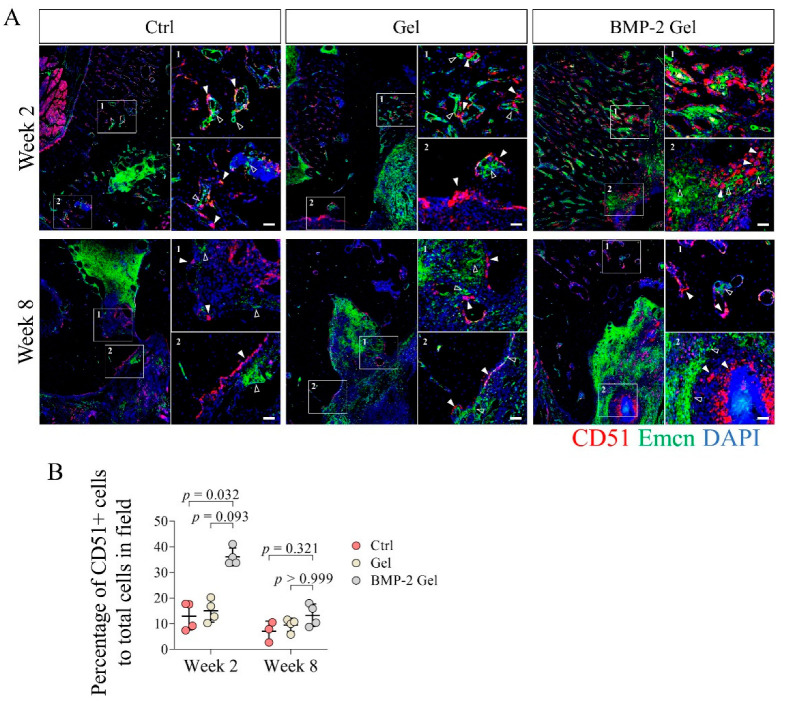
Implantation of BMP-2-loaded hydrogel facilitated the growth of CD51+ skeletal stem cells (SSCs) and their descendants during the bone regeneration of mandibular defects. (**A**) Immunofluorescent staining of CD51+ cells in the mandibular bone. Scale bar, 50 μm. The white arrow indicates the CD51+ SSCs and descendants. The hollow arrow indicates the vessels. (**B**) Quantitative analysis of the number of CD51+ cells in the new bone. The *p* value was calculated by one-way ANOVA and *Dunn’s* multiple comparisons test.

**Figure 6 pharmaceutics-14-02397-f006:**
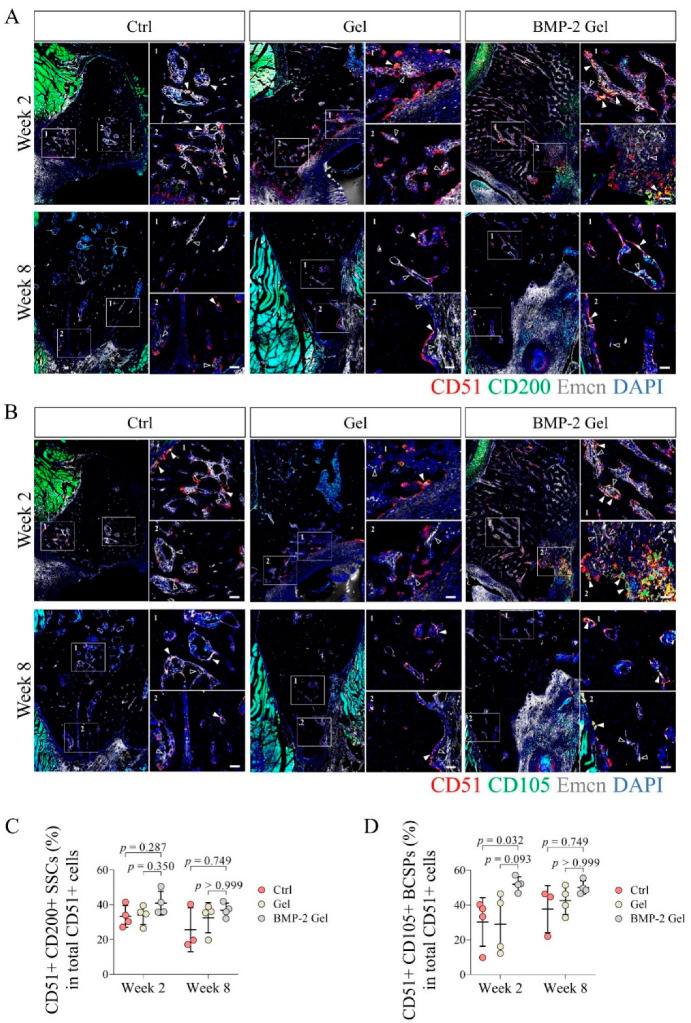
The fraction of CD51+ CD105+ SSCs descendants was elevated after implantation of BMP-2-loaded hydrogel in the newly regenerated bone. (**A**,**C**) Representative images and quantitative analysis of immunofluorescent staining of CD51+ CD200+ SSCs in the new bone. Scale bar, 50 μm. The white arrow indicates the CD51+ CD200+ SSCs. The hollow arrow indicates the vessels. (**B**,**D**) Representative images and quantitative analysis of immunofluorescent staining of CD51+ CD105+ SSCs descendants in the new bone. Scale bar, 50 μm. The white arrow indicates the CD51+ CD105+ SSCs descendants. The hollow arrow indicates the vessels. The *p* value was calculated by one-way ANOVA and *Dunn’s* multiple comparisons test.

## Data Availability

The data presented in this study are available upon contacting with the corresponding authors.

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
