# Peer review of "Controlled Release of Bone Morphogenetic Protein-2 Augments the Coupling of Angiogenesis and Osteogenesis for Accelerating Mandibular Defect Repair"

_pharmaceutics, 2022, doi:10.3390/pharmaceutics14112397_

Round 1

Reviewer 1 Report

This article is well-written and well organized. BMP-2 is known as growth factor and commercialized. So, in my thought, I recommend you to describe the specific point of your material in this article. For example, the strength of 'controlled release' or formation type of hydrogel must be described. 

Reviewer 2 Report

The study of Hao Yao et. al is aimed to bone regeneration in mandibular defects. As an alternative to the osseous free flaps transplantation used as standard method for reconstructing segmental mandibular defects, authors propose to apply a BMP-2 loaded hydrogel. As a model, a critical-size mandibular defect model of rat was used. Implantation of BMP-2 loaded hydrogel significantly promoted bone formation.

Regardless of the positive effect in bone formation, an important contribution of this study to the field would be the analysis at cellular level during the mandibular defect repair. Based in microscopy analysis, authors suggest that, like in mandibular development, healing of mandibular defect depends on the periosteum mediated intramembranous ossification. Also, the new bone formation induced by BMP-2 would be dependent of the type H vessels and associated osteoprogenitor cells. Thus, BMP-2 would be contributing to mandibular regeneration by coupling angiogenesis and osteogenesis.

 General Comments

 1. Please check the title. The data does no support that there is a “Controlled Release of Bone Morphogenetic Protein-2”.

 2. Please provide details about how the ratio of new bone to old bone was calculated (Fig. 2).

 3. In this study, important conclusions are obtained from immunofluorescence analysis. Therefore, data must be robust enough to draw conclusions:

- The quality of the images should be significantly improved (Fig 3-6). It may be that confocal microscopy or to present the not overlapping images could be useful.

- Fig. 3, How the area fraction of type H vessels in the new bone was calculated?

- Fig. 4-6, How the percent of cells was obtained?

- What was the criteria used for the selection of areas to analyze?

- For the analysis, how many visual fields or how many sequential sections were used?

 Specific Comments

Line 132. Check if the point followed corresponds.

Line 151. Check “FPA”. Is it PFA?

Reviewer 3 Report

The authors used BMP-2 loaded hydrogel to repair mandibular bone defects and discovered that BMP-2 not only promotes osteogenic differentiation but also promote angiogenesis, which has been reported in other studies. This results suggest as a bone growth factor, BMP-2 does not work on osteogenic differentiation stem cells. However, some concerns have been raised on some issues in the manuscript.

1. One of the most severe one is the titile. Authors declaimed they use controlled release of BMP-2 in their study while the kenetics of release of BMP-2 was not mentioned in the text at all.

2. The details of the calculation of sample size in the animal study should be supplemented.

3. "Hydrogel was synthesized by mixing above solution with 106 photo-initiator 2-hydroxy-4′-(2-hydroxyethoxy)-2-methylpropiophenone (Irgacure 2959, 107 Ciba) at a final concentration of 0.05% w/v followed by exposing to ultraviolet (UV, 95 108 mW/cm2) light for 6 minutes." Does the radiation with UV light damage bioactivity of BMP-2? Although the results of this study demonstrated after radiation, BMP-2 loaded hydrogel worked, this may imply UV light just did not damage all BMP-2. Will this fabrication of the hydrogel waste BMP-2? Is there any alternative synthetic method that is  milder and more friendly for BMP-2?

Round 2

Reviewer 2 Report

The authors have carefully reviewed and replied to all comments, providing new information where necessary. The text has been reviewed in detail and the conclusions have been appropriately weighted.